# Destructive effects of UVC radiation on *Drosophila melanogaster*: Mortality, fertility, mutations, and molecular mechanisms

Mohamed Lotfy[1]*, Aalaa Khattab[2], Mohammed Shata[1], Ahmad Alhasbani[1], Abdulla Almesmari[1], Saeed Alsaeedi[1], Saeed Alyassi[1], Biduth Kundu[1]

1 Biology Department, College of Science, United Arab Emirates University, Al Ain, United Arab Emirates,
2 Faculty of Dentistry, The British University in Egypt, El Sherouk City, Cairo, Egypt

* m.lotfy@uaeu.ac.ae

## Abstract

The detrimental effects of ultraviolet C (UVC) radiation on living organisms, with a specific focus on the fruit fly *Drosophila melanogaster*, were examined. This study investigated the impact of heightened UVC radiation exposure on *D. melanogaster* by assessing mortality and fertility rates, studying phenotypic mutations, and investigating the associated molecular mechanisms. The findings of this study revealed that UVC radiation increases mortality rates and decreases fertility rates in *D. melanogaster*. Additionally, phenotypic wing mutations were observed in the exposed flies. Furthermore, the study demonstrated that UVC radiation downregulates the expression of antioxidant genes, including superoxide dismutase (SOD), manganese-dependent superoxide dismutase (Mn-SOD), zinc-dependent superoxide dismutase (Cu-Zn-SOD), and the G protein-coupled receptor methuselah (MTH) gene. These results suggest that UVC radiation exerts a destructive effect on *D. melanogaster* by inducing oxidative stress, which is marked by the overexpression of harmful oxidative processes and a simultaneous reduction in antioxidant gene expression. In conclusion, this study underscores the critical importance of comprehending the deleterious effects of UVC radiation, not only to safeguard human health on Earth, but also to address the potential risks associated with space missions, such as the ongoing Emirate astronaut program.

## Introduction

Ultraviolet radiation (UVR) is a harmful solar radiation for life. UVR has the potential to inflict damage to cellular DNA, resulting in mutagenic abnormalities in humans and ultimately leading to skin cancer. Human exposure to solar UVR occurs daily and is amplified by the global degradation of the ozone stratospheric layer, which serves as a crucial protective shield against UVR.

Skin cancer is the most prevalent type of cancer, particularly among white populations in northern regions, where the prevalence and associated health burdens are notably high. Excessive exposure to UV radiation induces DNA mutations and is the primary driver of skin

**Funding:** This research received support from an SDG Grant (2022) awarded by Research Affairs at UAE University, Grant Code: G00004046. The funders had no role in study design, data collection and analysis, decision to publish, or preparation of the manuscript.

**Competing interests:** The authors have declared that no competing interests exist.

cancer. There are two common categories of skin cancer, melanoma and non-melanoma, both of which can lead to skin disfigurement and considerably affect the quality of life of patients with skin cancer [1].

Globally, approximately 130,000 new cases of melanoma are reported annually, with non-melanoma skin cancer accounting for approximately 3 million cases each year [2, 3]. According to a study conducted at Al Ain Hospital, the prevalence of skin cancer in the United Arab Emirates is 14.5% [4].

Sunlight emits a spectrum of electromagnetic radiation encompassing a range of wavelengths, including gamma rays, X-rays, ultraviolet (UV) rays, visible light, infrared radiation, microwaves, and radio waves. Ultraviolet radiation (UVR) is classified into three bands based on its wavelength: ultraviolet A (UVA; 315–400 nm), ultraviolet B (UVB; 280–315 nm), and ultraviolet C (UVC; 100–280 nm). The intensity of UVR reaching the Earth's surface varies owing to factors such as altitude, surface transparency, weather conditions, and the protective ozone stratospheric layer. The stratospheric ozone layer serves as a shield against harmful effects of UVR. It absorbs almost all UVC radiation, most UVB radiation, and a small portion of UVA radiation [5].

UVR exposure is the leading cause of photoaging and skin cancer, accounting for 65–90% of all melanomas [6]. The phototoxic mechanism of action of UVR induces gene mutations through the formation of DNA pyrimidine dimers or exacerbates oxidative DNA damage by generating excessive free radicals, thus promoting oxidative stress damage [7]. These free radicals typically result from certain metabolic reactions, and their production considerably increases upon overexposure to UVR [8].

UVC radiation is an exceptionally energetic and harmful form of UV radiation that adversely affects DNA. The interaction of UVC photons with DNA molecules can lead to the formation of pyrimidine dimers, DNA strand breaks, and generation of reactive oxygen species (ROS). This ROS production potentially causes DNA damage and alters DNA bases and sugar moieties. Additionally, this damage can disrupt normal cellular processes and trigger various cellular responses such as DNA repair mechanisms and cell cycle arrest. Failure to adequately repair this damage can result in mutations that contribute to the development of health issues, including skin cancers and genetic diseases. Therefore, protection against UVC radiation is of paramount importance to minimize its damaging effects on DNA and overall health [9–12].

The adverse effects of UVC radiation provide fertile ground for extensive biological research to understand how various cosmic factors affect the phenotypes and genotypes of the experimental animal model, *Drosophila melanogaster*. *D. melanogaster* is an exceptional animal model used in various studies because of its short and straightforward life cycle lasting approximately 12 days, allowing numerous successive generations to be observed within a few weeks [13]. Therefore, it is an ideal candidate for investigating the impact of UVC radiation on the lifespan and fertility of *D. melanogaster*. Furthermore, UVC exposure could potentially alter the normal physical structure of *D. melanogaster* owing to DNA damage induced by the overproduction of free radicals and diminished protective actions of antioxidants [14].

The human body continuously produces damaging free radical compounds along with protective antioxidant molecules aimed at counteracting their harmful effects. When the balance between these two elements is disrupted, an increase in free radical production and a decrease in antioxidant activity occur, a condition known as oxidative stress. This condition can lead to cellular DNA damage, tissue injury, organ dysfunction, and overall body failure, and is often associated with premature aging and various chronic and acute diseases [15].

G protein-coupled receptors (GPCRs) are cell-surface receptors involved in a wide array of cellular signaling processes. In *D. melanogaster*, methuselah (MTH) encodes a GPCR linked to

increased lifespan and stress resistance. Although direct studies linking DNA, oxidative stress, GPCRs, and the MTH are somewhat limited, oxidative stress has been shown to affect GPCR signaling, potentially influencing cellular responses, aging, and various illnesses [16–19].

The high prevalence of skin cancer in the UAE due to excessive sunlight exposure emphasizes the pressing need to raise awareness regarding UVC hazards to prevent radiation-induced diseases, particularly skin cancer. This has led to the development of vital strategies to reduce its incidence. These strategies include avoiding UVR exposure, offering guidance on skin cancer prevention, ensuring the accessibility of sun-safety resources, and advocating skin cancer screening programs [20].

Our study aligns with the objectives of the Emirate Astronaut Program, which seeks to send more Emirati astronauts into space on new missions, where they face a substantially higher UVC hazard than on Earth. Space research involving the International Space Station (ISS) aims to examine the diverse effects of prolonged exposure to space, including increased UVR levels, on living organisms [21, 22].

The primary objective of this study was to demonstrate the harmful effects of UVC radiation on skin cancers. Additionally, we aimed to elucidate the mutagenic mechanisms of UVC action by exploring the gene expression of the superoxide dismutase antioxidant enzyme and the G protein-coupled receptor methuselah at the molecular level.

## Material and methods

### *D. melanogaster* rearing and UVC treatment

Wild type *D. melanogaster* was obtained from the Carolina Biological Supply Company (Carolina, NC, USA). The flies were reared on a 12:12 light/dark cycle at 25°C in plastic containers and nourished with standard Carolina meal medium supplemented with yeast. Every 4 days, the flies were transferred to fresh medium.

For the UVC radiation treatment, 20 virgin females and 20 males of *D. melanogaster* were reared in new culture vials in groups of 40 flies per vial (n = 40), with each group represented in triplicates. The first group served as the non-irradiated control group (C). The second group was subjected to daily UVC irradiation for 1 minute over a period of seven consecutive days (UVC1). The third group received daily UVC irradiation for 2 minutes each day, also for seven days (UVC2). The fourth group was exposed to daily UVC irradiation for 3 minutes per day over seven days (UVC3).

After gently anesthetizing the flies, the culture vials were covered with Tehaux optical quartz plates (Amazon, China), maintaining a constant distance of 35 cm from a Ledvance UVC germicidal lamp with a power of 36 W and emitting radiation at a wavelength of 253.7 nm (Amazon, China). This daily exposure regimen was conducted consistently for seven days. Following the seven-day irradiation period, flies were transferred to new vials for subsequent experiments. By specifying both the duration of exposure per day and the number of days for each group, this revised description offers a comprehensive understanding of the UVC radiation exposure schedule for each experimental group [23–25].

### *D. melanogaster* mortality and fertility rate

The mortality rates in all four experimental groups were determined at the end of 14 days following the 7-day UVC irradiation period. To calculate the mortality rate, the number of dead flies in each group was counted. The formula for mortality rate is as follows: Mortality Rate (%) = (Number of Dead Flies / Total Initial Flies) × 100.

To estimate the fertility rate, 20 virgin males and 20 females were introduced into newly labeled vials, and mating was allowed for 7 days. Subsequently, all offspring were counted over

the following 14 days, starting with the emergence of the first adult from each vial. The formula for fertility rate is as follows: Fertility Rate (%) = (Number of Offspring / Total Initial Flies) × 100. These calculations were performed in accordance with established methods and procedures [26].

## *D. melanogaster* phenotypic mutations

As mentioned before, for the UVC radiation treatment, 20 virgin females and 20 males of *D. melanogaster* were reared in new culture vials in groups of 40 flies per vial (n = 40), with each group represented in triplicates. The first group served as the non-irradiated control group (C). The second group was subjected to daily UVC irradiation for 1 minute over a period of seven consecutive days (UVC1). The third group received daily UVC irradiation for 2 minutes each day, also for seven days (UVC2). The fourth group was exposed to daily UVC irradiation for 3 minutes per day over seven days (UVC3). After that from all groups 20 males and 20 females *D. melanogaster* progeny (first generation) were transferred to new vails. The phenotypic mutagenic mutations were examined flies for control group (C) and UVC1, UVC2, and UVC3 irradiation groups. Flies were anesthetized to detect their external structures and to observe any phenotypic and morphological changes. The percentage of *D. melanogaster* morphological and phenotypic mutations were calculated in each group. A SPOT digital camera (Diagnostic Instruments, Leica MZ6 microscope, Heerbrugg, Switzerland) was used to capture images of *D. melanogaster* morphology. All images were captured under uniform lighting conditions [13].

## *D. melanogaster* superoxide dismutase antioxidant

Male and female flies were divided into four groups as previously described in the *D. melanogaster* rearing and UVC treatment sub-section. After seven days of treatment, *D. melanogaster* flies from each group were homogenized in cold phosphate-buffered saline, centrifuged, and supernatants were collected to measure superoxide dismutase (SOD) levels and total protein. SOD activity was assessed by employing a readily available kit obtained from (Sigma-Aldrich, MI, USA), following the manufacturer's specified procedures. Our evaluation of superoxide dismutase activity utilized an indirect assay method centered around the activities of xanthine oxidase and a developed color reagent change. SOD's inhibitory effect was gauged by observing the reduction in color intensity at 440 nm that served as the basis for expressing SOD activity as a percentage inhibition rate. Furthermore, total protein concentration was determined using the quick start™ Bradford protein assay Kit, a commercially accessible product from (Bio-Rad, CA, USA). The Bradford assay relies on variations in absorbance because of the interaction between Coomassie Brilliant Blue dye and proteins, causing a shift from a reddish hue to a bluish one. The increase in absorbance at 595 nm was employed to quantify the protein concentration within the samples. SOD and total protein levels were calculated for each group as SOD-specific activity units per milligram of protein [27, 28].

## *D. melanogaster* gene expression via quantitative real-time PCR (qRT-PCR)

Male and female flies were divided into four groups, as previously described. After 7 days of treatments, *D. melanogaster* flies from each group were preserved in RNA later and stored at -80˚C. Total RNA was extracted from each of the four groups by homogenizing flies from each culture vessel using RNeasy Protect Kit (Qiagen, TX, USA). Subsequently, cDNA was synthesized using a high-capacity cDNA reverse transcription kit (Thermo Fisher Scientific, MA, USA). An Applied Biosystems QuantStudio-5 Real-Time PCR System was used for the qPCR

**Table 1. Primer sequences used for qRT-PCR.**

| Genes | Forward Primer (5'→3') | Reverse Primer (5'→3') |
|---|---|---|
| RpS20 | CCGCATCACCCTGACATCC | TGGTGATGCGAAGGGTCTTG |
| Cu-Zn-SOD | GCGGCGTTATTGGCATTG | ACTAACAGACCACAGGCTATG |
| Mn-SOD | CACATCAACCACACCATCTTC | CGTCTTCCACTGCGACTC |
| MTH | AGCGTATATTAGGAGTGAAGAAGG | CCGTAGGAAGAAGGTGTAAGTC |

analysis. A relative CT (CT) PowerUp SYBR Green detection method (Thermo Fisher Scientific) was employed with specific primers for the target genes (see Table 1). The average threshold cycle (Ct) values were used to determine the relative differences between the control UVC-non-irradiated and UVC-irradiated groups, with each sample normalized to RpS20 as the housekeeping reference gene. The Ct (threshold cycle) method was used to calculate the relative variations in gene expression using the $2^{-\Delta\Delta Ct}$ formula to estimate fold difference rates [29].

## Statistical analysis

The data were expressed as mean ± standard error of the mean. Statistical analysis was conducted using SPSS 15.0 software (IBM Corporation, Armonk, NY, USA), employing one-way ANOVA followed by post-hoc test to determine the significance differences between groups. A significance threshold of P less than 0.05 was set for all analyses. Asterisks (*) indicate significant differences of treated groups compared to the control group.

## Results

### Effect of UVC radiation on *D. melanogaster* mortality rate

The impact of UVC radiation on fertility rates in *D. melanogaster* across different experimental groups indicated that UVC radiation significantly adversely affected fertility rates in all UVC-irradiated groups compared to the normal, wild-type, and control groups (C) (Fig 1). Experimental groups exposed to UVC radiation exhibited reduced fertility rates, suggesting that UVC radiation negatively disrupts the reproductive capacity of flies. The reduction in fertility may be correlated with the duration of UVC exposure, as evidenced by the gradual decrease in fertility rates in the UVC1–UVC3 groups. These results suggest that UVC radiation exposure led to a notable increase in mortality rates across all treatment groups of *D. melanogaster* compared to the control group, indicating that UVC radiation has a damaging effect on the survival rate of UVC-irradiated flies.

### Effect of UVC radiation on *D. melanogaster* fertility rate

The impact of UVC radiation on fertility rates in *D. melanogaster* across different experimental groups showed that UVC irradiation had a significant adverse effect on fertility rates in all UVC-irradiated groups compared to the normal, wild-type, and non-irradiated control groups (C) (Fig 2). Experimental groups exposed to UVC radiation exhibited reduced fertility rates, suggesting that UVC radiation negatively suppresses the reproductive capacity of flies. The reduction in fertility may be correlated with the duration of UVC exposure, as seen by the gradually decreasing fertility rates in the UVC1–UVC3 groups. This underscores the detrimental effect of UVC irradiation on the reproductive success of these *D. melanogaster*-treated groups.

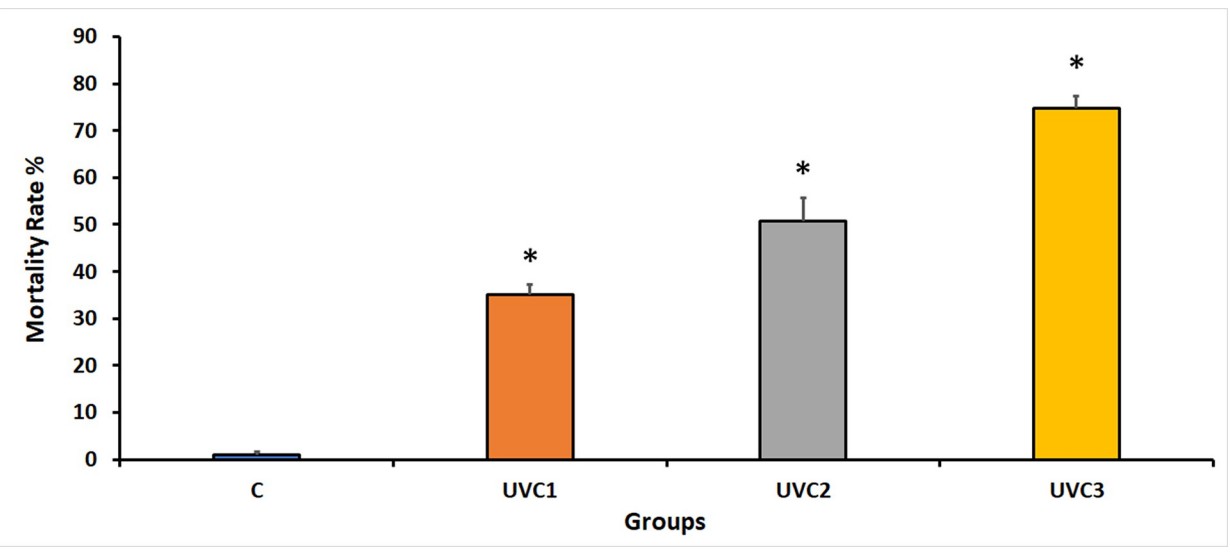

**Fig 1. *D. melanogaster* mortality rates in response to UVC irradiation.** This graph illustrates the mortality rates for various experimental groups of *D. melanogaster* in response to UVC radiation exposure. The experimental groups consisted of the normal, wild-type control group without UVC treatment (Group C), the group exposed to UVC radiation for 1 minute (UVC1), the group exposed to UVC radiation for 2 minutes (UVC2), and the group exposed to UVC radiation for 3 minutes (UVC3). UVC radiation led to a significant increase in mortality rates in all treated groups when compared to the untreated control group (Group C). Asterisks (*) indicate that the means of the treated groups were significantly different ($p<0.05$) compared to control group. The sample size for each group was 40 flies per vial, comprising 20 males and 20 females.

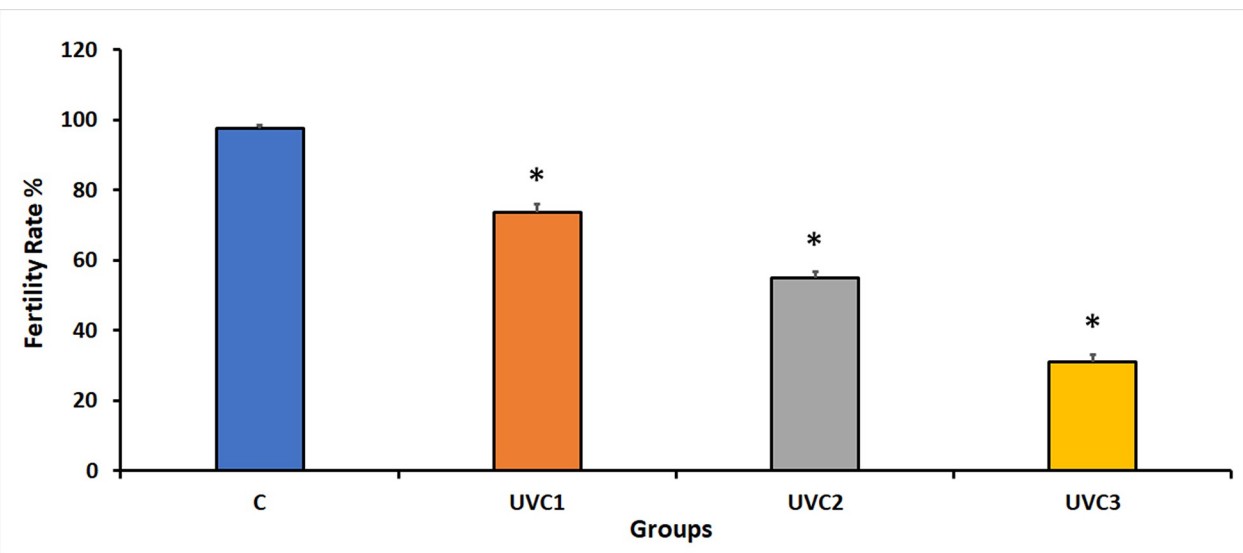

**Fig 2. Impact of UVC radiation on fertility rates in *D. melanogaster* on day 10.** This figure presents the average number of first-generation offspring (fertility rates) for various experimental groups of *D. melanogaster* on day 10. The experimental groups include the following: the normal, wild-type control group without UVC treatment (Group C), the group exposed to UVC radiation for 1 minute (UVC1), the group exposed to UVC radiation for 2 minutes (UVC2), and the group exposed to UVC radiation for 3 minutes (UVC3). UVC radiation had a significant detrimental effect on fertility, as it led to a substantial reduction in fertility rates in all UVC-irradiated groups when compared to the non-irradiated control group (Group C). The data is represented as the mean fertility rates ± standard error (SE). Asterisks (*) indicate that the means of the treated groups were significantly different ($p<0.05$) compared to control group. Each experimental group consisted of 40 flies per vial, with an equal distribution of 20 males and 20 females.

## Effect of UVC radiation on *D. melanogaster* phenotypic mutations

The effects of UVC radiation on the physical structure of *D. melanogaster*, specifically focusing on the phenotypic changes observed in the wing structure of the first generation, revealed that the UVC-treated groups (UVC1, UVC2, and UVC3) exhibited distinct wing mutations compared to the wild-type control groups (C) (Fig 3). These mutations include curved and crumpled wings in UVC1, vestigial and crumpled wings in UVC2, and cut wings in UVC3. This result underscores the experimental evidence for the impact of UVC irradiation on the physical structure of *D. melanogaster*, as evidenced by the distinctive wing mutations observed in

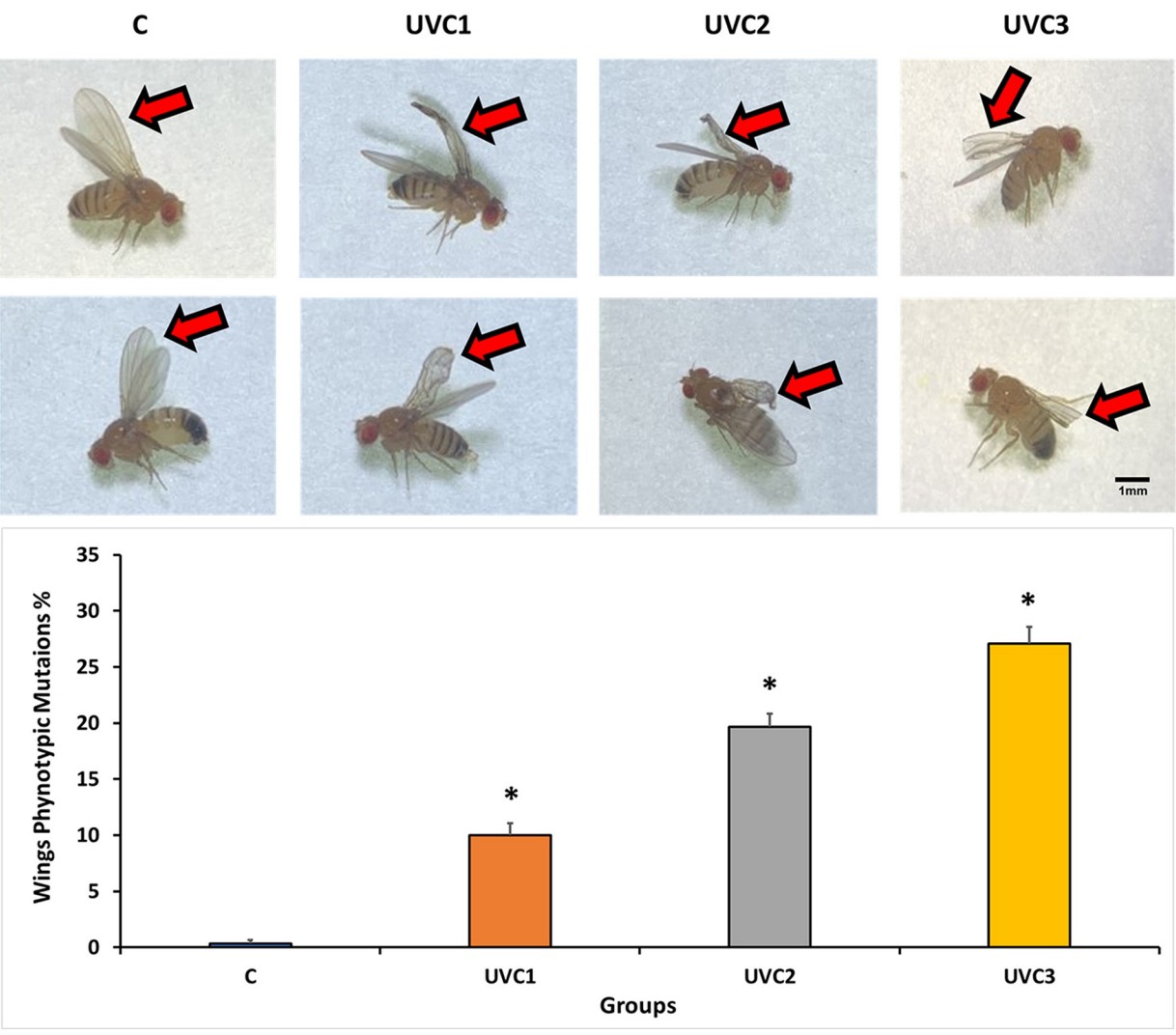

**Fig 3. Phenotypic effects of UVC radiation on *D. melanogaster*.** (A), images depict the henotypic consequences of UVC radiation exposure on *D. melanogaster* of the F1 generation. The experimental groups include the following: (C) Wild-type wings with no visible defects, (UVC1) wings displaying curved and crumpled mutations, (UVC2) wings exhibiting vestigial and crumpled mutations, and (UVC3) wings demonstrating cut mutations. The *D. melanogaster* groups are categorized as follows: the normal, wild-type control group without UVC treatment (Group C), the group exposed to UVC radiation for 1 minute (UVC1), the group exposed to UVC radiation for 2 minutes (UVC2), and the group exposed to UVC radiation for 3 minutes (UVC3). Observation and imaging of *D. melanogaster* were conducted under a dissecting microscope. Each experimental group consisted of 40 flies per vial, comprising 20 males and 20 females. Scale bar = 1 mm. (B), the quantification of the results showed the percentage of different wings phenotypic mutations. The data is represented as the mean wings mutations percentage ± standard error (SE). Asterisks (*) indicate that the means of the treated groups were significantly different (p<0.05) compared to control group. Each experimental group consisted of 40 flies per vial, with an equal distribution of 20 males and 20 females.

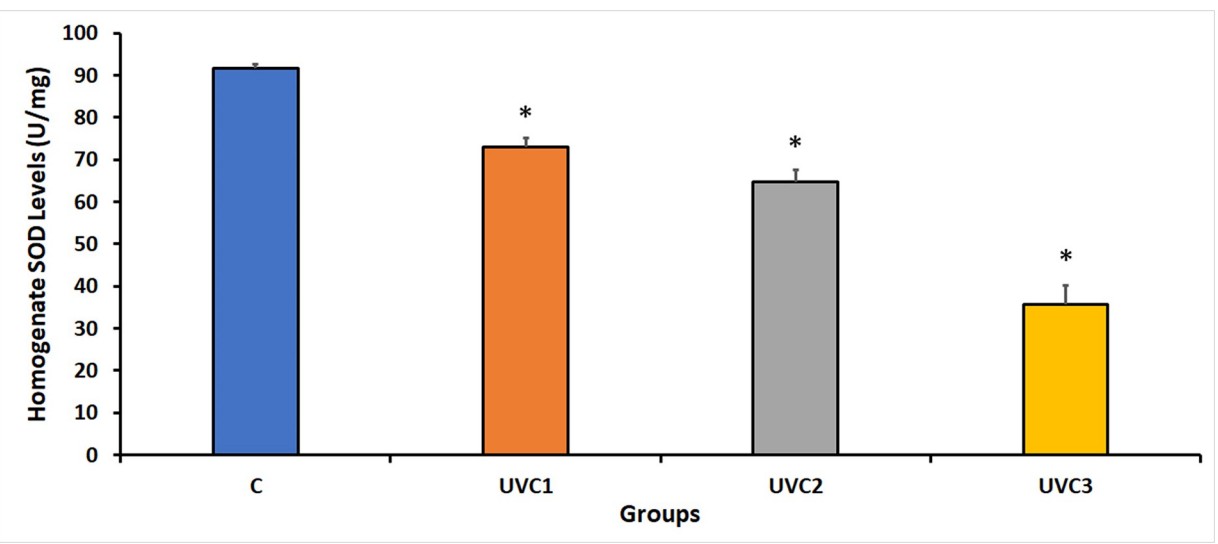

**Fig 4. Superoxide dismutase (SOD) activity in response to UVC radiation on *D. melanogaster*.** This figure displays the levels of superoxide dismutase (SOD) activity for different experimental groups of *D. melanogaster*. The experimental groups comprise the following: the non-treated control group (C) without UVC exposure, the group irradiated with UVC for 1 minute (UVC1), the group irradiated with UVC for 2 minutes (UVC2), and the group irradiated with UVC for 3 minutes (UVC3). Notably, UVC radiation led to a significant reduction in SOD activity in all UVC-irradiated groups when compared to the control group (C). Asterisks (*) indicate that the means of the treated groups were significantly different (p<0.05) compared to control group. Each experimental group comprised 40 flies per vial, with an equal distribution of 20 males and 20 females.

the treated groups. The dose-dependent relationship between radiation duration and phenotypic changes supports these results. In addition, a quantification of the percentage of different wings phenotypic mutations in each group showed that there was an increment in UVC radiation dose dependent manner. This effect explores the mutagenic impact of UVC radiation on some phenotypic traits in *D. melanogaster*.

## Effect of UVC radiation on *D. melanogaster* superoxide dismutase levels

The results of the experiment focusing on the levels of SOD homogenate activity, an enzyme involved in combating oxidative stress, in different experimental groups of *D. melanogaster* revealed that UVC irradiation had a considerable impact on the inhibition of SOD activity in all UVC-irradiated groups compared to the control groups (C) (Fig 4). This suggests that UVC irradiation negatively affects the antioxidant defense system of flies. The reduction in SOD activity may be linked to the duration of UVC exposure, as indicated by the decrease in SOD levels in the UVC1–UVC3 groups. This indicated the negative impact of UVC irradiation on the antioxidant defense mechanisms of the treated flies.

## Effect of UVC radiation on *D. melanogaster* Mn-SOD gene expression

The results of the expression of the Mn-SOD gene in *D. melanogaster* under UVC irradiation showed a reduction in the expression of the Mn-SOD gene in all UVC-irradiated groups compared to the normal, wild-type, and non-irradiated control groups (C) (Fig 5). This finding suggests that UVC radiation destructively affects the expression of Mn-SOD, which may be influenced by the duration of UVC exposure, as indicated by the significant reduction in Mn-SOD expression in the UVC2 group. This highlights the harmful effects of UVC radiation on the expression of the Mn-SOD antioxidant enzyme.

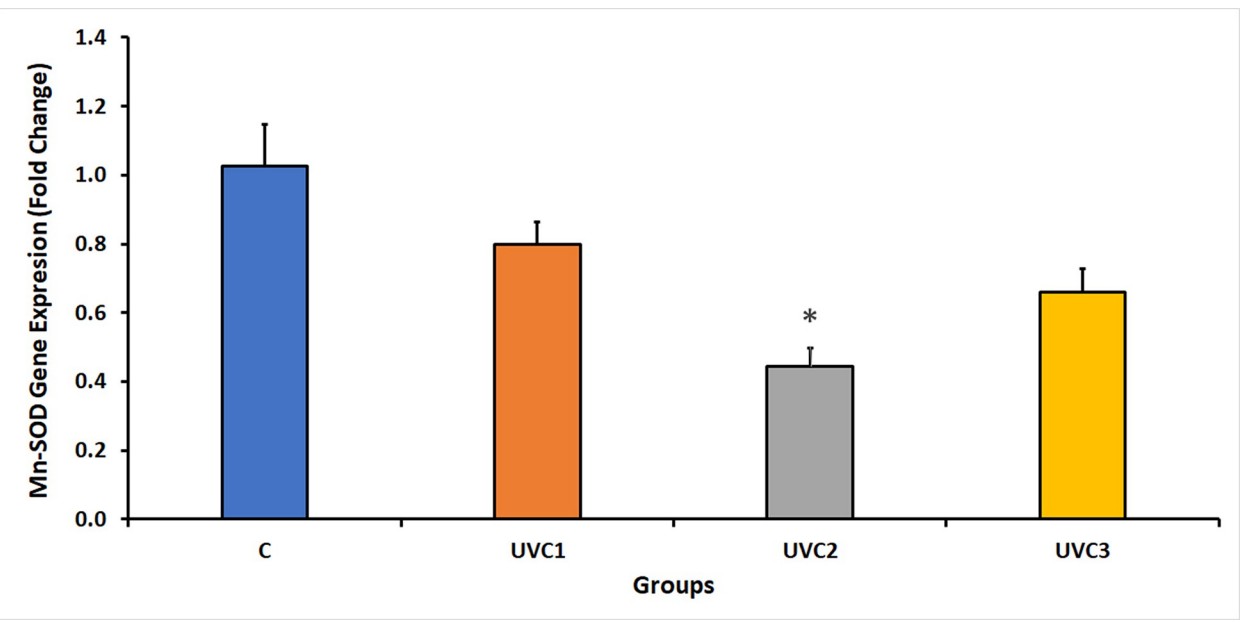

**Fig 5. Fold change in *D. melanogaster* manganese-dependent superoxide dismutase (Mn-SOD) expression in response to UVC radiation.** This figure illustrates the fold change in Mn-SOD expression in *D. melanogaster* in response to ultraviolet-C (UVC) radiation. The fold change was determined through quantitative real-time PCR, with the housekeeping gene RpS20 serving as a reference. Significantly, UVC radiation resulted in a marked reduction in Mn-SOD expression in all UVC-irradiated groups when compared to the control group (C). The *D. melanogaster* groups included the following: the normal, untreated control group without UVC exposure (Group C), the group exposed to UVC radiation for 1 minute (UVC1), the group exposed to UVC radiation for 2 minutes (UVC2), and the group exposed to UVC radiation for 3 minutes (UVC3). Asterisks (*) indicate that the means of the treated groups were significantly different ($p < 0.05$) compared to control group. Each experimental group comprised 40 flies per vial, with an equal distribution of 20 males and 20 females. (See S1 Raw data, showing the Mn-SOD qPCR raw data).

### Effect of UVC radiation on *D. melanogaster* Cu-Zn-SOD gene expression

The UVC radiation results showed a significant inhibition of Cu-Zn-SOD gene expression in *D. melanogaster* in all UVC-irradiated groups compared to that in the control groups (C) (Fig 6). A reduction in Cu-Zn-SOD gene expression indicated that UVC radiation negatively affected the expression of the Cu-Zn-SOD antioxidant enzyme, which might be influenced by the duration of UVC exposure, with a decrease in gene expression as the exposure time increased. This highlights the damaging effect of UVC radiation on the expression of Cu-Zn-SOD.

### Effect of UVC radiation on *D. melanogaster* MTH expression

UVC irradiation of *D. melanogaster* showed significant suppression of the expression of the MTH in all UVC-irradiated groups compared to the control groups (C) (Fig 7). This suggests that UVC irradiation adversely affects the expression of MTH, a G protein-coupled receptor gene that can be influenced by the duration of UVC exposure, as evidenced by a decrease in MTH gene expression levels with increasing exposure time.

## Discussion

Ultraviolet (UV) radiation, including UVC radiation, is a component of the electromagnetic spectrum with distinct implications for biological systems. Despite being naturally blocked by the ozone layer, UVC can be artificially generated in laboratories [30]. Investigations involving UVC radiation and *D. melanogaster* have contributed considerably to our understanding of

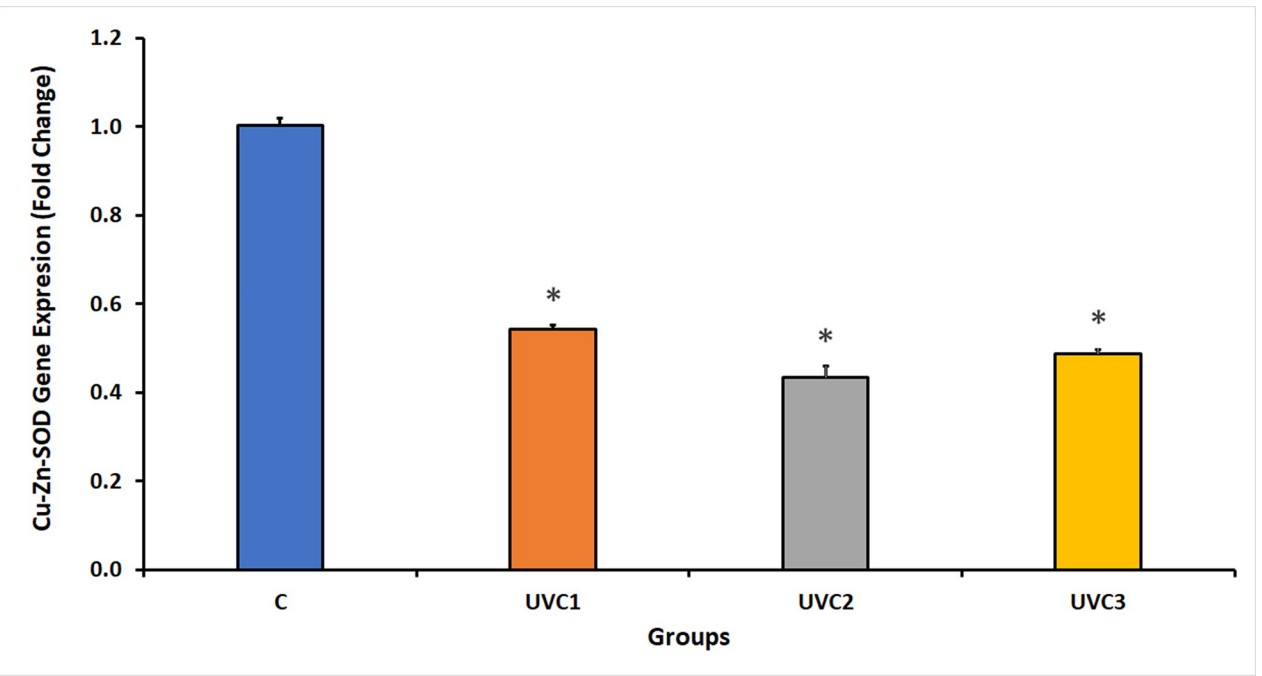

**Fig 6. Fold change in *D. melanogaster* Copper-Zinc-Dependent Superoxide Dismutase (Cu-Zn-SOD) expression in response to UVC radiation.**
This figure portrays the fold change in Cu-Zn-SOD expression in *D. melanogaster* under the influence of ultraviolet-C (UVC) radiation. Quantitative real-time PCR was employed to determine the fold change, using the housekeeping gene RpS20 as a reference. Notably, UVC radiation induced a substantial reduction in Cu-Zn-SOD expression in all UVC-irradiated groups when compared to the control group (C). The *D. melanogaster* groups consisted of the following: the normal, untreated control group without UVC exposure (Group C), the group exposed to UVC radiation for 1 minute (UVC1), the group exposed to UVC radiation for 2 minutes (UVC2), and the group exposed to UVC radiation for 3 minutes (UVC3). Asterisks (*) indicate that the means of the treated groups were significantly different (p<0.05) compared to control group. Each experimental group was composed of 40 flies per vial, with an equal distribution of 20 males and 20 females. (See S2 Raw data, showing the Cu-Zn-SOD qPCR raw data).

DNA repair mechanisms, cell cycle regulation, apoptosis, and the overall impact of UVC radiation on organismal health. While UVA and UVB radiation from sunlight affect *D. melanogaster* in its natural habitat, the effects of UVC radiation are less prevalent owing to atmospheric shielding [25].

*D. melanogaster*, commonly known as the fruit fly, is a valuable model organism in scientific research owing to its short lifespan, ease of maintenance, and biological similarities to higher organisms. Fruit flies exhibit various biological and behavioral responses to controlled doses of UVC radiation. UVC radiation, known for its high mutagenic potential, can directly inflict DNA damage, thereby giving rise to genetic mutations and cellular abnormalities. UVC exposure induces DNA damage and oxidative stress and disrupts cellular functions in fruit flies [31–35]. This fundamental mechanism of action may explain the elevated mortality rate and diminished fertility observed in the UVC-exposed groups of *D. melanogaster* in the current study.

The adverse consequences of UVC radiation on *D. melanogaster* include a reduced lifespan, diminished fertility, and the appearance of phenotypic mutations. In females, UVC radiation impedes egg production and increases egg mortality, whereas it hampers sperm production and viability [36]. The mechanisms underlying these effects involve DNA damage and disruption of reproductive processes. The accumulation of UVC-induced DNA damage and compromised cellular functions contribute to accelerated aging and increased mortality [37]. This underlying mechanism of action may explain the documented increase in mortality rate and

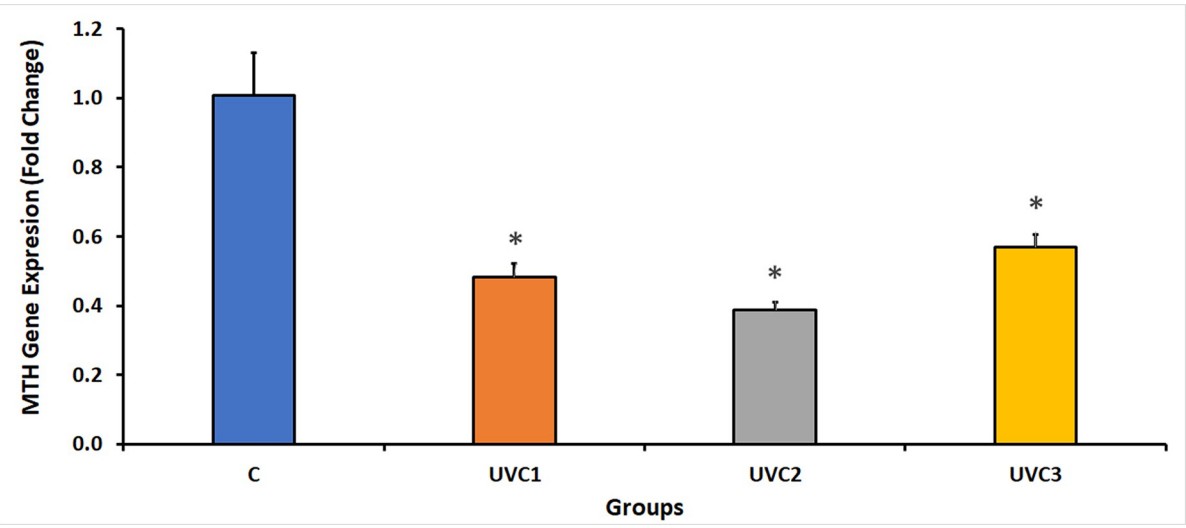

**Fig 7. Fold change in *D. melanogaster* G protein-coupled receptor gene, methuselah (MTH), expression under UVC radiation.** This figure illustrates the fold change in MTH gene expression in *D. melanogaster* exposed to ultraviolet-C (UVC) radiation. Quantitative real-time PCR was employed to assess the fold change, utilizing the housekeeping gene RpS20 as a reference. Remarkably, UVC radiation led to a substantial reduction in MTH gene expression in all UVC-irradiated groups when compared to the control group (C). The *D. melanogaster* groups consisted of the following: the normal, untreated control group without UVC exposure (Group C), the group exposed to UVC radiation for 1 minute (UVC1), the group exposed to UVC radiation for 2 minutes (UVC2), and the group exposed to UVC radiation for 3 minutes (UVC3). Asterisks (*) indicate that the means of the treated groups were significantly different ($p<0.05$) compared to control group. Each experimental group comprised 40 flies per vial, evenly divided between 20 males and 20 females. (See S3 Raw data, showing the MTH qPCR raw data).

decrease in fertility rate observed in the UVC-treated *D. melanogaster* groups in the current study.

UVC radiation directly damages DNA, resulting in the formation of DNA abrasions such as cyclobutane pyrimidine dimers and 6–4 photoproducts. These lesions impede DNA replication and transcription, thereby disrupting normal cellular processes [38]. Exposure to UVC during specific developmental stages can lead to developmental defects and abnormalities that affect pupation, metamorphosis, and adult emergence. These abnormalities may include developmental delays and malformations, ultimately influencing the overall viability of the flies [39]. These effects were consistent with those observed in our study, which recorded certain wing morphological mutations in *D. melanogaster* subjected to UVC irradiation.

Superoxide dismutase (SOD) enzymes, including Cu-Zn-SOD and Mn-SOD, play pivotal roles in protecting cells against oxidative stress by converting superoxide radicals into less deleterious compounds. In *D. melanogaster*, UVC radiation considerably diminishes the activity of antioxidant enzymes, resulting in increased oxidative stress and cellular damage [40]. This advantageous protective function of SOD isoform gene expression was attenuated in the current study following UVC irradiation of *D. melanogaster*. This suggests that UVC radiation amplifies oxidative stress, leading to perturbations in *D. melanogaster* tissues, thereby contributing to an augmented mortality rate and reduced fertility rate associated with the accumulation of free radicals, as postulated in a prior study [41].

The methuselah (MTH) gene in *D. melanogaster* encodes a G protein-coupled receptor (GPCR) that exerts a substantial influence on the regulation of fruit fly lifespan. Overexpression of MTH in *D. melanogaster* has been associated with extended longevity and is linked to various physiological processes, including stress response, immunity, and resistance to oxidative stress [42, 43]. In the current study, UVC irradiation diminished the expression of MTH,

which may have contributed to a shortened lifespan, increased mortality rate, and inhibition of the fertility of *D. melanogaster*. This seemingly contradictory outcome could be explained by the fact that UVC radiation simultaneously reduces MTH expression while exerting deleterious effects on DNA mutations and the overall dysfunction of 'body systems of *D. melanogaster*, including the reproductive system [13]. Additionally, the expression of genes associated with oxidative stress resistance, such as MTH, was suppressed under UVC radiation, further exacerbating the observed detrimental effects. In addition, a viable approach within the scope of gene radioresistance involves the augmentation of endogenous antioxidants. Notably, the upregulation of specific antioxidative entities, such as Mn-SOD, emerges as a strategic intervention. Mn-SOD, functioning as a potent scavenger, effectively counteracts the reactive oxygen species generated after the cellular response to radiation [44].

The expression of Mn-SOD, Cu-Zn-SOD and MTH in Figs 5–7 respectively showed decrease in UVC1 and UVC2 groups but suddenly increased at the UVC 3 group could be explained according to two recent publications on 2022 [45, 46], as the rapid gene expression fluctuations on a minute timescale are due to critical element of a cell's response to a range of stressors, with their magnitude dependent on the stress's duration, onset speed, and intensity. These intricate regulatory mechanisms function at various levels, including transcriptional, post-transcriptional, and post-translational control, facilitating swift adaptation and recovery from stress. Specifically, post-transcriptional processes play a central role by enabling the rapid adjustment of RNA turnover, translational activity, covalent modifications, and subcellular RNA localization through a network of cellular pathways. This dynamic gene expression regulation allows cells to efficiently confront environmental and pathophysiological challenges, reprogramming their molecular functions to maintain protein quality control. When stressors are alleviated or removed, these minute-scale fluctuations ensure a quick return to normal cellular operations, safeguarding protein homeostasis. This underscores the pivotal role of post-transcriptional responses, with their rapidity measured in minutes, in the cell's ability to swiftly adapt and survive during transient stress episodes, providing insight into how organisms withstand sudden environmental changes while preserving their internal cellular conditions for growth, functionality, and overall survival.

These reports indicate that *D. melanogaster* will be able to overcome the initial inhibitory response to gene expression after UV radiation and recover quickly. Moreover, the above-mentioned reports showed that the inhibition-increment cycle within minutes scale in gene expression observed in *D. melanogaster* is indeed feasible and it is related to the defensive mechanism of the organism.

## Conclusion

The detrimental effects of UVC radiation on *D. melanogaster*, the focus of this study, yield valuable insights into the impact of UVC radiation on a thoroughly investigated model organism. This choice allowed for a more precise and comprehensive analysis of these effects. This study delved beyond the mortality and fertility rates to encompass phenotypic mutations, providing a holistic understanding of how UVC exposure influences various aspects of *D. melanogaster* biology. An investigation into the molecular mechanisms underlying the deleterious effects of UVC radiation on *D. melanogaster* adds depth to the research, including the downregulation of antioxidant genes such as SOD and MTH, thereby providing a clearer depiction of the biological pathways involved.

Establishing a link between UVC-induced damage and oxidative stress provides a mechanistic rationale for the observed effects, shedding light on the potential underlying causes and opening avenues for further improvement strategies. The implications of this study extend

beyond *D. melanogaster* for human health. Understanding the destructive effects of UVC radiation can lead to measures aimed at safeguarding humans both on Earth and during space missions, underscoring the relevance of these findings for real-world applications. Mentioning the potential impact of UVC radiation on astronauts, including those participating in programs such as the Emirate Astronaut program, introduces a futuristic dimension to the significance of the study, emphasizing the applicability of the research in practical scenarios.

The conclusion of this study underscores the necessity of comprehending the detrimental effects of UVC radiation to formulate protective measures, imbuing a forward-looking perspective that underscores the potential benefits of proactively addressing the issue. Exploring the downregulation of antioxidant genes expands our understanding of how UVC radiation affects biological systems, with potential implications beyond *D. melanogaster* which could contribute to broader research on the effects of UVC radiation.

The inclusion of phenotypic mutations in the analysis offers a more comprehensive view of the impact of UVC radiation, potentially yielding insights into the evolutionary processes and adaptive responses to UVC radiation. By combining multiple levels of analysis, ranging from mortality and fertility rates to molecular mechanisms and gene expression, the current study adopted a holistic approach to understand the effects of UVC radiation on *D. melanogaster*, thereby contributing to a nuanced understanding of this phenomenon. This study presents a multifaceted and innovative approach to investigate the detrimental effects of UVC radiation on *D. melanogaster*, making a valuable contribution to the field of research.

In summary, UVC radiation has deleterious effects on *D. melanogaster* by diminishing lifespan, reducing fertility, and causing the emergence of mutagenic phenotypic mutations. In addition, UVC irradiation suppressed the expression of various SOD and MTH genes. Our results revealed that the adverse effects of UVC radiation may primarily result from oxidative stress-induced damage associated with the reduced expression of antioxidant genes, resulting in increased mortality, reduced fertility, and the appearance of morphological mutations in *D. melanogaster*. Ultimately, investigating the destructive effects of UVC radiation on *D. melanogaster* will enhance our awareness of the damaging effects of UVC radiation, whether on Earth or in space, as exemplified by the ongoing Emirate astronaut program.

## Supporting information

**S1 Raw data.**
(XLSX)

**S2 Raw data.**
(XLSX)

**S3 Raw data.**
(XLSX)

**S1 File.**
(DOCX)

**S2 File.**
(DOCX)

## Acknowledgments

My full acknowledgments to Research Affairs at UAE University and Biology Department support.

## Author Contributions

**Conceptualization:** Mohamed Lotfy.

**Data curation:** Mohamed Lotfy, Aalaa Khattab, Mohammed Shata, Ahmad Alhasbani, Abdulla Almesmari, Saeed Alsaeedi, Saeed Alyassi, Biduth Kundu.

**Formal analysis:** Mohamed Lotfy, Aalaa Khattab, Mohammed Shata, Ahmad Alhasbani, Abdulla Almesmari, Saeed Alsaeedi, Saeed Alyassi.

**Funding acquisition:** Mohamed Lotfy.

**Investigation:** Mohamed Lotfy, Mohammed Shata, Ahmad Alhasbani, Saeed Alsaeedi, Saeed Alyassi.

**Methodology:** Mohamed Lotfy, Mohammed Shata, Ahmad Alhasbani, Abdulla Almesmari, Saeed Alsaeedi, Saeed Alyassi, Biduth Kundu.

**Project administration:** Mohamed Lotfy.

**Resources:** Mohamed Lotfy, Biduth Kundu.

**Software:** Mohamed Lotfy, Aalaa Khattab, Mohammed Shata, Ahmad Alhasbani, Abdulla Almesmari, Saeed Alsaeedi, Saeed Alyassi, Biduth Kundu.

**Supervision:** Mohamed Lotfy.

**Validation:** Mohamed Lotfy, Mohammed Shata, Ahmad Alhasbani, Abdulla Almesmari, Saeed Alsaeedi.

**Visualization:** Mohamed Lotfy, Mohammed Shata, Ahmad Alhasbani, Abdulla Almesmari, Saeed Alsaeedi, Saeed Alyassi.

**Writing – original draft:** Mohamed Lotfy, Aalaa Khattab, Mohammed Shata, Ahmad Alhasbani, Abdulla Almesmari, Saeed Alsaeedi, Saeed Alyassi.

**Writing – review & editing:** Mohamed Lotfy.

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
