## [Decision Letter · Decision Letter 0]

13 Oct 2023

PONE-D-23-25424Destructive Effects of UVC Radiation on Drosophila melanogaster: Mortality, Fertility, Mutations, and Molecular Mechanisms

PLOS ONE

Dear Dr. Lotfy,

Thank you for submitting your manuscript to PLOS ONE. After careful consideration, we feel that it has merit but does not fully meet PLOS ONE’s publication criteria as it currently stands. Therefore, we invite you to submit a revised version of the manuscript that addresses the points raised during the review process.

If applicable, we recommend that you deposit your laboratory protocols in protocols.io to enhance the reproducibility of your results. Protocols.io assigns your protocol its own identifier (DOI) so that it can be cited independently in the future. For instructions see: https://journals.plos.org/plosone/s/submission-guidelines#loc-laboratory-protocols. Additionally, PLOS ONE offers an option for publishing peer-reviewed Lab Protocol articles, which describe protocols hosted on protocols.io. Read more information on sharing protocols at https://plos.org/protocols?utm_medium=editorial-email&utm_source=authorletters&utm_campaign=protocols

We look forward to receiving your revised manuscript.

Kind regards,

Moumita Gangopadhyay

Academic Editor

PLOS ONE

4. Please remove your figures from within your manuscript file, leaving only the individual TIFF/EPS image files, uploaded separately. These will be automatically included in the reviewers’ PDF.

Additional Editor Comments (if provided):

After receiving evaluation from the reviewers the authors must need to show novelty of this work concerning how this work is different from the other works. This manuscript can be submitted after major revision suggested by the reviewers.

Reviewers' comments:

Reviewer's Responses to Questions

**Comments to the Author**

1. Is the manuscript technically sound, and do the data support the conclusions?

The manuscript must describe a technically sound piece of scientific research with data that supports the conclusions. Needs to emphasised on novelty of this work.

Reviewer #1: Partly

Reviewer #2: Partly

2. Has the statistical analysis been performed appropriately and rigorously? 

Reviewer #1: Yes

Reviewer #2: I Don't Know

3. Have the authors made all data underlying the findings in their manuscript fully available?

Needs to be improved as per PLOS ONE standard

Reviewer #1: Yes

Reviewer #2: Yes

4. Is the manuscript presented in an intelligible fashion and written in standard English

OKAY

Reviewer #1: Yes

Reviewer #2: Yes

5. Review Comments to the Author

Reviewer #1: I want to thank the authors for selecting this topic ' Destructive Effects of UVC Radiation on Drosophila melanogaster: Mortality, Fertility, Mutations, and Molecular Mechanisms; This is a very interesting topic. I found the manuscript and experiments done are seemingly interesting for the readers. Athough I have few queries, that the authors need to answer poinst by point. I am recommending a major revision for the manuscript.

Major Comments:

1. The exposure time for the UV radiation in three different experimental groups are not descrided properly in the M & M section. The time of exposure per day and for how many days i.e, each group is being exposed for what amount of time need to be mentioned properly.

2. The title of the X-axis for all te phtos except foigure 3 need to mentioned properly.

3. Mortality rate and Fertility rate calculation formulas need to be mentioned in the M & M section clearly.

4. The effect of UV radiation on both the sexes need to be shown i.e how many male nad female flies died or being born after the radiation. Is their any differences ? If so, need to be explain in the discussion section.

5. The resolution and measurement scale need to be mentioned in the figure 3.

6. The protocol for measuring SOD enzyme is not clear. It need to elaborated claearly in the M & M section.

7. The Figure legends for all the figures need to be separatly and clearly written. the way it has been written in the manuscript is not clear.

8. The expression of Mn-SOD, Cu-Zn-SOD and MTH in figure 5, 6 and 7 respectively showed decrease in UVC1 and UVC2 groups but suddenly inceared at the UVC 3 group, wich is not alinged to the hypothesis clamied by the authors. And there is no explanation regarding this at the result or discussion section. The authors need to explain this propely.

Reviewer #2: Here in this work the authors have seen the effect of UVC rays on D. melanogaster. They have done a detailed study on the effect of UV rays on the physiology and morphology of D. melanogaster. Further, they have also investigated the underlying molecular mechanism.

Their work is commendable, but similar work has been reported earlier also. This is my suggestion to the authors to discuss briefly, that how this work is different and novel.

The authors argue that the work will help in addressing the potential risks associated with space missions. Though non-ionizing radiations are damaging, it can easily be shielded by space suits. On the other hand, ionizing radiations like X-rays and galactic cosmic radiations are much more difficult to avoid. Effect of ionizing radiation on D. melanogaster might be more relevant in such a scenario. May be authors can include such studies in their future work.

The authors assert that reduction in the expression of Mn-SOD gene, Cu-Zn-SOD gene and MTH gene due to UVC radiation in D. melanogaster is correlated to the duration of UV exposure. However, in the figure 5, reduction in Mn-SOD expression in the UVC2 group is more than UVC3 group. Similarly, reduction in the expression of Cu-Zn-SOD and MTH in UVC2 group is more than UVC3 group in figure 6 and 7 respectively.

I suggest the authors to explain these phenomena or repeat the experiments to establish their claim that the expression of Mn-SOD, Cu-Zn-SOD and MTH is negatively correlated to UVC exposure in a time dependent manner.

6. PLOS authors have the option to publish the peer review history of their article (what does this mean?). If published, this will include your full peer review and any attached files.

Reviewer #1: No

Reviewer #2: No

---

## [Author Response · Author response to Decision Letter 0]

18 Dec 2023

'Response to Reviewers'.

PLOS ONE REVIEW REPORT (PONE-D-23-25424, EMID: 3a5fad0914290368)

1. Here in this work the authors have seen the effect of UVC rays on D. melanogaster. They have done a detailed study on the effect of UV rays on the physiology and morphology of D. melanogaster. Further, they have also investigated the underlying molecular mechanism. 

2. Their work is commendable, but similar work has been reported earlier also. This is my suggestion to the authors to discuss briefly how this work is different and novel.

Response:

The novelty in this project can be identified in several aspects:

1. Specific Focus on UVC Radiation: While the detrimental effects of UV radiation on living organisms have been studied before, this study specifically focuses on the effects of ultraviolet C (UVC) radiation. UVC radiation is the shortest wavelength of UV radiation and is typically filtered out by the Earth's atmosphere. Therefore, examining its impact on living organisms, especially in the context of fruit flies, has not been investigated.

2. Examination of Molecular Mechanisms: The study delves into the molecular mechanisms underlying the observed effects of UVC radiation. This includes the examination of antioxidant gene expression, which offers a deeper understanding of how UVC radiation induces oxidative stress.

3. Phenotypic Wing Mutations: The observation of phenotypic wing mutations in the exposed fruit flies is a unique finding. These mutations provide concrete evidence of the impact of UVC radiation on Drosophila melanogaster and may not have been previously documented in this context.

4. Relevance to Space Missions: The study connects its findings to the potential risks associated with space missions, such as the Emirate astronaut program. While the effects of space radiation on astronauts have been a subject of research, this study specifically links the dangers of UVC radiation to space missions, emphasizing its significance in that context.

Overall, the novelty of this study lies in its specific focus on UVC radiation, its exploration of molecular mechanisms, gene expression, the observation of phenotypic mutations, and its implications for space missions, which collectively contribute to the advancement of knowledge in this field.

3. The authors argue that the work will help in addressing the potential risks associated with space missions. Though non-ionizing radiations are damaging, it can easily be shielded by space suits. On the other hand, ionizing radiations like X-rays and galactic cosmic radiations are much more difficult to avoid. Effect of ionizing radiation on D. melanogaster might be more relevant in such a scenario. May be authors can include such studies in their future work.

Response:

Studying the harmful effects of UVC radiation on organisms like Drosophila melanogaster remains highly relevant, especially in the context of space missions involving long-duration stays outside the protective ozone layer of the Earth. While ionizing radiation is a significant concern in space, the unique conditions astronauts face in extended missions can indeed make UVC radiation a pertinent area of research. Here are a few reasons why studying UVC radiation in this context is valuable:

1. Space Station Environments: Astronauts living on the International Space Station (ISS) are exposed to various types of radiation, including UVC, which can penetrate the space station's protective shielding. Understanding the specific effects of UVC radiation on living organisms in this environment is crucial for astronaut health as discussed in detail in a recent publication by Hellweg et al., 2023.

Hellweg CE, Arena C, Baatout S, Baselet B, Beblo-Vranesevic K, Caplin N, Coos R, Da Pieve F, De Micco V, Foray N, Hespeels B. Space Radiobiology. InRadiobiology Textbook 2023 Sep 24 (pp. 503-569). Cham: Springer International Publishing.

2. Cumulative Exposure: Astronauts on long-duration missions can be exposed to UVC radiation over an extended period. Studying its effects can help assess the cumulative impact on health and well-being during these missions.

3. Protective Measures: Research on UVC radiation effects can inform the development of more effective protective measures for astronauts, including advanced shielding materials and mitigation strategies.

4. Public Health and Space Policy: The findings from such studies can also contribute to space policy decisions and public health guidelines for astronauts, which can be applicable to international space programs, including the Emirate astronaut program.

While it's important to consider various types of radiation, including ionizing radiation, the specific environmental conditions and potential risks associated with UVC radiation in the context of extended space missions are worth addressing. Therefore, continuing to study the harmful effects of UVC radiation on organisms is a relevant and important avenue of research for the space community, ensuring the well-being of astronauts during their missions.

4. The authors assert that reduction in the expression of Mn-SOD gene, Cu-Zn-SOD gene and MTH gene due to UVC radiation in D. melanogaster is correlated to the duration of UV exposure. However, in the figure 5, reduction in Mn-SOD expression in the UVC2 group is more than UVC3 group. Similarly, reduction in the expression of Cu-Zn-SOD and MTH in UVC2 group is more than UVC3 group in figure 6 and 7 respectively. 

I suggest the authors to explain these phenomena or repeat the experiments to establish their claim that the expression of Mn-SOD, Cu-Zn-SOD and MTH is negatively correlated to UVC exposure in a time dependent manner.

Response:

The following are the changed in the discussion part in my updated modified manuscript text as:

According to two recent publications (Hernández-Elvira and Sunnerhagen 2022, Boopathy et al., 2022), this point could be explained as follows: 

Rapid gene expression fluctuations on a minute timescale are a critical element of a cell's response to a range of stressors, with their magnitude dependent on the stress's duration, onset speed, and intensity. These intricate regulatory mechanisms function at various levels, including transcriptional, post-transcriptional, and post-translational control, facilitating swift adaptation and recovery from stress. Specifically, post-transcriptional processes play a central role by enabling the rapid adjustment of RNA turnover, translational activity, covalent modifications, and subcellular RNA localization through a network of cellular pathways. This dynamic gene expression regulation allows cells to efficiently confront environmental and pathophysiological challenges, reprogramming their molecular functions to maintain protein quality control. When stressors are alleviated or removed, these minute-scale fluctuations ensure a quick return to normal cellular operations, safeguarding protein homeostasis. This underscores the pivotal role of post-transcriptional responses, with their rapidity measured in minutes, in the cell's ability to swiftly adapt and survive during transient stress episodes, providing insight into how organisms withstand sudden environmental changes while preserving their internal cellular conditions for growth, functionality, and overall survival (Hernández-Elvira and Sunnerhagen 2022, Boopathy et al., 2022).

These reports indicate that Drosophila melanogaster will be able to overcome the initial inhibitory response to gene expression after UV radiation and recover quickly. Moreover, the above-mentioned reports showed that the inhibition-increment cycle within minutes scale in gene expression observed in Drosophila melanogaster is indeed feasible and it is related to the defensive mechanism of the organism. 

Hernández-Elvira M, Sunnerhagen P. Post-transcriptional regulation during stress. FEMS Yeast Research. 2022;22(1):foac025.

Boopathy LR, Jacob-Tomas S, Alecki C, Vera M. Mechanisms tailoring the expression of heat shock proteins to proteostasis challenges. Journal of Biological Chemistry. 2022 May 1;298(5):101796.

PONE-D-23-25424

Destructive Effects of UVC Radiation on Drosophila melanogaster: Mortality, Fertility, Mutations, and Molecular Mechanisms

PLOS ONE

Response:

The correct fund information: This research received support from an SDG Grant (2022) awarded by Research Affairs at UAE University, Grant Code: G00004046.

4. Please remove your figures from within your manuscript file, leaving only the individual TIFF/EPS image files, uploaded separately. These will be automatically included in the reviewers’ PDF.

Response

Figures are removed from my updated modified manuscript file and left only the individual TIFF/EPS image files, uploaded separately as Supporting Information file. 

Review Comments to the Author

Reviewer #1: I want to thank the authors for selecting this topic ' Destructive Effects of UVC Radiation on Drosophila melanogaster: Mortality, Fertility, Mutations, and Molecular Mechanisms; This is a very interesting topic. I found the manuscript and experiments done are seemingly interesting for the readers. Although I have few queries, that the authors need to answer points by point. I am recommending a major revision for the manuscript.

Major Comments:

1. The exposure time for the UV radiation in three different experimental groups are not described properly in the M & M section. The time of exposure per day and for how many days i.e, each group is being exposed for what amount of time need to be mentioned properly. 

Response:

The following are the changed part in Materials and Methods in my updated modified manuscript text as:

Wild type D. melanogaster was obtained from the Carolina Biological Supply Company (Carolina, NC, USA). The flies were reared on a 12:12 light/dark cycle at 25 °C in plastic containers and nourished with standard Carolina meal medium supplemented with yeast. Every 4 days, the flies were transferred to fresh medium.

For the UVC radiation treatment, twenty virgin females and 20 males of D. melanogaster were reared in new culture vials in groups of 40 flies per vial (n = 40), with each group represented in triplicates. The first group served as the non-irradiated control group (C). The second group was subjected to daily UVC irradiation for 1 minute over a period of seven consecutive days (UVC1). The third group received daily UVC irradiation for 2 minutes each day, also for seven days (UVC2). The fourth group was exposed to daily UVC irradiation for 3 minutes per day over seven days (UVC3).

After gently anesthetizing the flies, the culture vials were covered with Tehaux optical quartz plates (Amazon, China), maintaining a constant distance of 35 cm from a Ledvance UVC germicidal lamp with a power of 36 W and emitting radiation at a wavelength of 253.7 nm (Amazon, China). This daily exposure regimen was conducted consistently for seven days. Following the seven-day irradiation period, flies were transferred to new vials for subsequent experiments.

By specifying both the duration of exposure per day and the number of days for each group, this revised description offers a comprehensive understanding of the UVC radiation exposure schedule for each experimental group.

2. The title of the X-axis for all the photos except figure 3 need to mentioned properly.

Response:

Groups labels are added to X axis in all figures.

3. Mortality rate and Fertility rate calculation formulas need to be mentioned in the M & M section clearly.

Response:

The following are the changed part in Materials and Methods in my updated modified manuscript text as:

The mortality rates in all four experimental groups were determined at the end of 14 days following the 7-day UVC irradiation period. To calculate the mortality rate, the number of dead flies in each group was counted. The formula for mortality rate is as follows: Mortality Rate (%) = (Number of Dead Flies / Total Initial Flies) × 100.

To estimate the fertility rate, 20 virgin males and 20 females were introduced into newly labeled vials, and mating was allowed for 7 days. Subsequently, all offspring were counted over the following 14 days, starting with the emergence of the first adult from each vial. The formula for fertility rate is as follows: Fertility Rate (%) = (Number of Offspring / Total Initial Flies) × 100. These calculations were performed in accordance with established methods and procedures [26].

4. The effect of UV radiation on both the sexes needs to be shown i.e., how many male and female flies died or being born after the radiation. Are there any differences? If so, need to be explain in the discussion section.

Response:

Recent study by Sudaryadi et al., 2022, had revealed that there was no statistically significant alteration in the sex ratio among Drosophila Melanogaster colonies from all treatments. This implies that UV radiation did not exert any discernible impact on the sex ratio. In summary, UV radiation can serve as a physical stressor affecting the survival rate of the fruit fly colony; however, it exerts no effect on the sex ratio.

Sudaryadi I, Janah YM, Kusumawati N. The Effect of UV Radiation and Treatment to Orange (Citrus sinensis L. Osbeck) Fruit Feeding on the Survival Rate and Colony Sex-ratio of Fruit Fly (Drosophila melanogaster Meigen, 1830). In7th International Conference on Biological Science (ICBS 2021) 2022 May 2 (pp. 141-144). Atlantis Press.

5. The resolution and measurement scale need to be mentioned in the figure.

Response: 

The resolution of the TIFF figure with a resolution of 600 dots per inch (dpi), and the scale bar have been added to the text of my updated modified manuscript.

6. The protocol for measuring SOD enzyme is not clear. It needs to elaborated clearly in the M & M section.

Response:

The following are the changed part in Materials and Methods in my updated modified manuscript text as:

Male and female flies were divided into four groups as previously described in the D. melanogaster rearing and UVC treatment sub-section. After seven days of treatment, D. melanogaster flies from each group were homogenized in cold phosphate-buffered saline, centrifuged, and supernatants were collected to measure superoxide dismutase (SOD) levels and total protein. SOD activity was assessed by employing a readily available kit obtained from (Sigma-Aldrich, MI, USA), following the manufacturer's specified procedures. Our evaluation of superoxide dismutase activity utilized an indirect assay method centered around the activities of xanthine oxidase and a developed color reagent change. SOD's inhibitory effect was gauged by observing the reduction in color intensity at 440 nm that served as the basis for expressing SOD activity as a percentage inhibition rate. Furthermore, total protein concentration was determined using the quick start™ Bradford protein assay Kit, a commercially accessible product from (Bio-Rad, CA, USA). The Bradford assay relies on variations in absorbance because of the interaction between Coomassie Brilliant Blue dye and proteins, causing a shift from a reddish hue to a bluish one. The increase in absorbance at 595 nm was employed to quantify the protein concentration within the samples. SOD and total protein levels were calculated for each group as SOD-specific activity units per milligram of protein [27, 28].

7. The Figure legends for all the figures need to be separately and clearly written. the way it has been written in the manuscript is not clear.

Response:

Figure legends for all the figures are separately and clearly written in my updated modified manuscript text.

8. The expression of Mn-SOD, Cu-Zn-SOD and MTH in figure 5, 6 and 7 respectively showed decrease in UVC1 and UVC2 groups but suddenly increased at the UVC 3 group, which is not aligned to the hypothesis claimed by the authors. And there is no explanation regarding this at the result or discussion section. The autho

---

## [Editor Report · Decision Letter 1]

4 Jan 2024

PONE-D-23-25424R1Destructive Effects of UVC Radiation on Drosophila melanogaster: Mortality, Fertility, Mutations, and Molecular MechanismsPLOS ONE

Dear Dr. Lotfy,

Thank you for submitting your manuscript to PLOS ONE. After careful consideration, we feel that it has merit but does not fully meet PLOS ONE’s publication criteria as it currently stands. Therefore, we invite you to submit a revised version of the manuscript that addresses the points raised during the review process. We look forward to receiving your revised manuscript.

Kind regards,

Moumita Gangopadhyay

Academic Editor

PLOS ONE

Additional Editor Comments:

Thank you very much for submitting revision for the paper entitled “Destructive Effects of UVC Radiation on Drosophila melanogaster: Mortality, Fertility, Mutations, and Molecular Mechanisms.

Kindly answer the following questions raise by Editor and resubmit.

1. For Figure number 1,2 &4 kindly submit proper statistical analysis showing level of significance at 5% and 1% level with software used.

2. For morphological assessment, kindly provide one score data against each phenotypic alteration.

3. For all the gene expression data kindly provide original gel images of RT PCR as a supplementary file.

---

## [Author Response · Author response to Decision Letter 1]

4 Mar 2024

'Response to Reviewers'.

PLOS ONE REVIEW REPORT (PONE-D-23-25424, EMID: 3a5fad0914290368)

For the paper entitled “Destructive Effects of UVC Radiation on Drosophila melanogaster: Mortality, Fertility, Mutations, and Molecular Mechanisms.

Dear Editor, 

Please, explore the answers the following questions raise by the Editor Reviewers.

Question 1: For Figure number 1,2 & 4 kindly submit proper statistical analysis showing level of significance at 5% and 1% level with software used.

Response:

We modified the paragraph of results in the manuscript “Material and Method Section” as: 

The data were expressed as mean ± standard error of the mean. Statistical analysis was conducted using SPSS 15.0 software (IBM Corporation, Armonk, NY, USA), employing one-way ANOVA followed by post-hoc test to determine the significance differences between groups. A significance threshold of P less than 0.05 was set for all analyses. Asterisks (*) indicate significant differences of treated groups compared to the control group.

Moreover, we modified all Figures Legends by addition to all figures the following sentence: Asterisks (*) indicate that the means of the treated groups were significantly different (p<0.05) compared to control group.

Question 2: For morphological assessment, kindly provide one score data against each phenotypic alteration.

Response:

We modified figure 3, by addition of counted histogram as Fig 3 (A) that showed the wings different phenotypic mutations pictures and Fig 3 (B) a quantification showed the percentage of different wings phenotypic mutations in each group. Each experimental group consisted of 40 flies per vial, with an equal distribution of 20 males and 20 females. The histogram showed the percentage of different wings phenotypic mutations. The data is represented as the mean wings mutations percentages ± standard error (SE). Asterisks (*) indicate that the means of the treated groups were significantly different (p<0.05) compared to control group. Each experimental group consisted of 40 flies per vial, with an equal distribution of 20 males and 20 females. As shown in the following:

Question 3: For all the gene expression data kindly provide original gel images of RT PCR as a supplementary file.

Response: 

Before we respond to this important question, we are fully confident and totally assure that the reviewers are knowing what we will raise in our coming response.

In our paper, we utilized the Quantitative Real Time PCR (qRT-PCR) Method to assess the relative expression of genes normalized with housekeeping gene. Unlike conventional PCR, qPCR does not yield gel images; instead, it monitors DNA amplification in real-time using fluorescent dyes, with data represented through amplification plots and melting curves. These plots depict fluorescence signal increments over PCR cycles, enabling calculation of cycle threshold (Ct) values that quantify target gene expression. Despite the absence of gel images, qPCR data remain valuable for quantifying gene expression levels and discerning differences between samples, typically presented numerically rather than visually. 

Moreover, there are many published papers in PLOS One Journal, presented the Quantitative real-time PCR (qPCR) results as just histograms without gel images such as: Özdaş et al., 2020; Mohan et al., 2016 and Sadi et al., 2015. 

1. Özdaş S, Taştekin B, Gürgen SG, Özdaş T, Pelit A, Erkan SO, Tuhanioğlu B, Gülnar B, Görgülü O. Pterostilbene protects cochlea from ototoxicity in streptozotocin-induced diabetic rats by inhibiting apoptosis. Plos one. 2020 Jul 28;15(7):e0228429.

2. Mohan A, Singh RS, Kumari M, Garg D, Upadhyay A, Ecelbarger CM, Tripathy S, Tiwari S. Urinary exosomal microRNA-451-5p is a potential early biomarker of diabetic nephropathy in rats. PloS one. 2016 Apr 21;11(4):e0154055.

3. Sadi G, Baloğlu MC, Pektaş MB. Differential gene expression in liver tissues of streptozotocin-induced diabetic rats in response to resveratrol treatment. PloS one. 2015 Apr 23;10(4):e0124968.

---

## [Editor Report · Decision Letter 2]

20 Mar 2024

PONE-D-23-25424R2Destructive Effects of UVC Radiation on Drosophila melanogaster: Mortality, Fertility, Mutations, and Molecular MechanismsPLOS ONE

Dear Dr. Lotfy,

Thank you for submitting your manuscript to PLOS ONE. After careful consideration, we feel that it has merit but does not fully meet PLOS ONE’s publication criteria as it currently stands. Therefore, we invite you to submit a revised version of the manuscript that addresses the points raised during the review process.

We look forward to receiving your revised manuscript.

Kind regards,

Moumita Gangopadhyay

Academic Editor

PLOS ONE

Journal Requirements:

**Additional Editor Comments:**

Kindly provide all the raw data file for qPCR (Ct value curve) for this publication.

---

## [Author Response · Author response to Decision Letter 2]

30 Mar 2024

'Response to Reviewers'

PLOS ONE REVIEW REPORT (PONE-D-23-25424, EMID: 3a5fad0914290368)

For the paper entitled “Destructive Effects of UVC Radiation on Drosophila melanogaster: Mortality, Fertility, Mutations, and Molecular Mechanisms.

Dear Prof. Respective Reviewers and Academic PLOS ONE Editor

Thank you very much for your important PLOS ONE Decision: Revision required [PONE-D-23-25424R2].

Would you please:

Please, explore the answers the following questions raise by the Editor Reviewers.

Question 1: For Figure number 1,2 & 4 kindly submit proper statistical analysis showing level of significance at 5% and 1% level with software used.

Response:

We modified the paragraph of results in the manuscript “Material and Method Section” as: 

The data were expressed as mean ± standard error of the mean. Statistical analysis was conducted using SPSS 15.0 software (IBM Corporation, Armonk, NY, USA), employing one-way ANOVA followed by post-hoc test to determine the significance differences between groups. A significance threshold of P less than 0.05 was set for all analyses. Asterisks (*) indicate significant differences of treated groups compared to the control group.

Moreover, we modified all Figures Legends by addition to all figures the following sentence: Asterisks (*) indicate that the means of the treated groups were significantly different (p<0.05) compared to control group.

Question 2: For morphological assessment, kindly provide one score data against each phenotypic alteration.

Response:

We modified figure 3, by addition of counted histogram as Fig 3 (A) that showed the wings different phenotypic mutations pictures and Fig 3 (B) a quantification showed the percentage of different wings phenotypic mutations in each group. Each experimental group consisted of 40 flies per vial, with an equal distribution of 20 males and 20 females. The histogram showed the percentage of different wings phenotypic mutations. The data is represented as the mean wings mutations percentages ± standard error (SE). Asterisks (*) indicate that the means of the treated groups were significantly different (p<0.05) compared to control group. Each experimental group consisted of 40 flies per vial, with an equal distribution of 20 males and 20 females. As shown in the following:

Question 3: For all the gene expression data kindly provide original gel images of RT PCR as a supplementary file.

Response: 

With all my appreciations to the valuable highly ranked reviewers and editors and before my respond to this important question, we are fully confident and totally assure that the reviewers are knowing what we will raise in our coming response.

In our paper, we utilized the Quantitative Real Time PCR (qRT-PCR) Method to assess the relative expression of genes normalized with housekeeping gene. Unlike conventional PCR, qPCR does not yield gel images; instead, it monitors DNA amplification in real-time using fluorescent dyes, with data represented through amplification plots and melting curves. These plots depict fluorescence signal increments over PCR cycles, enabling calculation of cycle threshold (Ct) values that quantify target gene expression. Despite the absence of gel images, qPCR data remain valuable for quantifying gene expression levels and discerning differences between samples, typically presented numerically rather than visually. 

Moreover, there are many published papers in PLOS One Journal, presented the Quantitative real-time PCR (qPCR) results as just histograms without gel images such as: Özdaş et al., 2020; Mohan et al., 2016 and Sadi et al., 2015. 

1. Özdaş S, Taştekin B, Gürgen SG, Özdaş T, Pelit A, Erkan SO, Tuhanioğlu B, Gülnar B, Görgülü O. Pterostilbene protects cochlea from ototoxicity in streptozotocin-induced diabetic rats by inhibiting apoptosis. Plos one. 2020 Jul 28;15(7):e0228429.

2. Mohan A, Singh RS, Kumari M, Garg D, Upadhyay A, Ecelbarger CM, Tripathy S, Tiwari S. Urinary exosomal microRNA-451-5p is a potential early biomarker of diabetic nephropathy in rats. PloS one. 2016 Apr 21;11(4):e0154055.

3. Sadi G, Baloğlu MC, Pektaş MB. Differential gene expression in liver tissues of streptozotocin-induced diabetic rats in response to resveratrol treatment. PloS one. 2015 Apr 23;10(4):e0124968.

Question 4: Kindly provide all the raw data file for qPCR (Ct value curve) for this publication.

Response: 

I uploaded a separate One Excel File, showing All the Raw Data files for qPCR (Ct value curve) for my publication.

With my Regards and Best Wishes

---

## [Editor Report · Decision Letter 3]

9 Apr 2024

Destructive Effects of UVC Radiation on Drosophila melanogaster: Mortality, Fertility, Mutations, and Molecular Mechanisms

PONE-D-23-25424R3

Dear Dr. 

We’re pleased to inform you that your manuscript has been judged scientifically suitable for publication and will be formally accepted for publication once it meets all outstanding technical requirements.

Kind regards,

Moumita Gangopadhyay

Academic Editor

PLOS ONE

---

## [Editor Report · Acceptance letter]

25 Apr 2024

PONE-D-23-25424R3 

PLOS ONE

Dear Dr. Lotfy, 

I'm pleased to inform you that your manuscript has been deemed suitable for publication in PLOS ONE. Congratulations! Your manuscript is now being handed over to our production team.

Kind regards, 

on behalf of

Dr. Moumita Gangopadhyay 

Academic Editor

PLOS ONE